# Sedation Rate Reduction in Paediatric Renal Nuclear Medicine Examinations: Consequences of a Targeted Audit

**DOI:** 10.3390/children8050424

**Published:** 2021-05-20

**Authors:** Christa Gernhold, Nina Kundtner, Martin Steinmair, Martin Henkel, Josef Oswald, Bernhard Haid

**Affiliations:** 1Department of Pediatric Urology, Hospital of the Sisters of Charity, Seilerstätte 4, 4020 Linz, Austria; nina.kundtner@gmx.at (N.K.); josef.oswald@ordensklinikum.at (J.O.); bernhard.haid@me.com (B.H.); 2Department of Nuclear Medicine, Hospital of the Sisters of Charity, Seilerstätte 4, 4020 Linz, Austria; martin.steinmair@ordensklinikum.at; 3Department of Paediatrics, Hospital of the Sisters of Charity Seilerstätte 4, 4020 Linz, Austria; martin.henkel@ordensklinikum.at

**Keywords:** children, nuclear medicine, sedation

## Abstract

Background: Nuclear medicine investigations are essential diagnostic tools in paediatric urology. Child-orientated examination techniques and the avoidance of sedation or anaesthesia vary in different institutions. We aimed at evaluating child friendly measures in our department to identify the potential for improvement. Based on these data, we changed the standards regarding the sedation policy and consequently re-evaluated sedation rates. Methods: Four-hundred thirty-five consecutive investigations were evaluated regarding the need for sedation, outcome and patient satisfaction at our department. After the revision of our department standards, we re-evaluated 159 examinations. Statistical analysis was performed with JUMBO (Java-supported Münsterian biometrical platform). Results: Eighty-six percent (60/70) would agree to perform an investigation under identical conditions again. Seventy-seven percent (17/22) of eligible patients >5 years of age felt good during the investigation. By changing our sedation policy, we could reduce the sedation rate from 27.1% to 7.5% (*p* < 0.0001; OR 0.219 95% CI 0.111–0.423). Conclusion: The evaluation of child friendly examination protocols demonstrated high reliability and patient satisfaction using situational sedation with a relatively high proportion of patients being sedated. Through protocol adaption with clear age limits, individual indication and education of staff, as well as the use of optimized sedatives, the need for sedation could be further reduced whilst maintaining a high patient satisfaction.

## 1. Introduction

Nuclear medicine investigations like 99mTc-mercaptoacetyltriglycin (MAG3) and 99mTc-dimercaptosuccinic acid (DMSA) scans play an essential role in the evaluation of congenital malformations of the urogenital system [1,2]. With these malformations being among the commonest, there is a high frequency of these examinations being performed. The Society of Nuclear medicine Procedure Guidelines clearly recommends “child friendly conditions” avoiding sedation or general anaesthesia by the implementation of an attentive and caring approach towards the children [3].

While some studies addressed the technical aspects of the examination conditions, the impact of the circumstances of those examinations on patients and their families with respect to discomfort and anxiety, however, is poorly investigated. To avoid the anticipated distress of undergoing nuclear medicine investigations, several examination conditions have shown a beneficial impact on decreasing anxiety: Parents offer the best of comfort and security for the child and should stay with the child during the whole procedure. Distraction strategies including toys, children’s books and TV or video goggles with child-oriented films can help to calm the child. The vast majority of children can be adequately immobilized using simple techniques such as vacuum cushions [3,4].

Not only do distraction of the children and good preparation of the family lead to a better acceptance of the examination, the nuclear medicine staff and the spatial adaptation to children also make an important contribution to a relaxed examination [5].

Nevertheless, the use of intravenous sedation or general anaesthesia and i.v. sedation, as well as the generalized use of oral or rectal sedatives (e.g., chloralhydrate, midazolam) are common. This not only exposes children to unnecessary medication, but also might confer an inhibition for clinicians to indicate the exams as warranted.

Kai et al. described the prevailing condition of sedation for carrying out nuclear medicine examinations in Denmark in children under three years of age. Based on their data, they were able to show that changing the sedation modality did not lead to an increase in the duration of the examination [6].

To quantify the national approach, a survey of all nuclear medicine departments in Austria was conducted (Part 1). The protocol in use at our department was evaluated retrospectively over a period of one year. Furthermore, patient and parental satisfaction, as well as nuclear medicine staff’s attitude toward paediatric nuclear medicine examinations were evaluated by a prospective survey (Part 2).

With the knowledge gained, the protocol was revised with targeted advice to staff and parents aiming at further reduction of the sedation rate. 

Consequently, after three years, an audit was carried out, and the outcomes were compared to the initial analysis (Part 3).

We hypothesized that despite a lower sedation rate, the implementation of a child friendly environment with an optimized protocol would yield satisfying outcomes.

## 2. Patients and Methods

Approval from the institutional ethics committee (EK 09/15), as well as informed consent from all prospectively included participants were obtained.

### 2.1. Standard Process for Nuclear Medicine Examinations at Our Department

Prior to the examination and irrespective of planned sedation, normal fluid and solid food intake without fasting were allowed. To guarantee adequate hydration, all patients received standardized intravenous hydration (“ELO-Paed balanced with 1% glucose” electrolyte solution by Fresenius Kabi, Germany).

For patients undergoing an MAG3 investigation, the infusion was started one hour before study initiation and for the DMSA scan, 30 min before the administration of the radioactive tracer. The individual tracer dose was calculated for each child according to the recommendations of the European Association of Nuclear Medicine (EANM) [4]. Tracer dosage was calculated based on an institutional standard operating procedure based on the EANM dose recommendations.

The basic renogram consisted of a dynamic phase with a minimum duration of 30 min for F + 20, a post micturition image after a mean of 45 min and, in cases with <50% activity on the post micturition image, a late image 2 h post injection.

For the DMSA scintigraphy, we not only performed planar imaging, but also SPECT (renal single-photon emission computerized tomography (SPECT)) imaging to guarantee a better detection of cortical defects; therefore, the entire DMSA scintigraphy took about 45 min.

The MAG3 and DMSA scans were performed using a Siemens Symbia T2 SPECT camera (Siemens Medical Solutions USA, Hoffman Estates, IL, USA).

In order to attain optimal quality, the children had to lie in a supine position for 40 to 60 min. The parents were informed in detail about the process and encouraged to prepare the children—as far as age-adapted possible—for the examination situation. Importantly, especially younger children had a meal beforehand. All children were accompanied by a trusted person, mostly parents with the exception of pregnant mothers. Weight- and age-adapted vacuum mattresses were used in babies and infants to reduce motion artefacts. The parents were present throughout the whole investigation to comfort and calm their children. Toys, children’s books and a TV with child-oriented films were available. The nuclear medicine staff were available for additional support in comforting the children if needed.

### 2.2. Part 1

In order to assess the current practice in nuclear examinations of children in all isotope departments of Austria performing nuclear medicine exams of infant kidney, *n* = 26 were contacted by phone and email.

### 2.3. Part 2

During a prospective study phase (February 2015–May 2015), parents of all children undergoing MAG3 renography or DMSA scintigraphy were included in the study and asked to complete a questionnaire.

Patients aged over 5 years were questioned using a visual analogue scale with face pictograms to differentiate mood. The questionnaire included questions about the wellbeing of the child (example: How did you feel during the examination? Would you come again?) and parents’ assessment of the children’s satisfaction as used in younger children (example: Do you find the examination situation very stressful? Was your child calm during the examination? Did you feel well looked after?)

In addition, the nuclear medicine staff’s opinion was assessed by a further questionnaire.

Additionally, a retrospective review of nuclear medicine examinations for one year (January 2014–May 2015) was performed, to assess the age-related use of sedatives, as well as the rate of unsuccessful exams.

During this time, most children between the ages of one and three and those appearing unrestful, as well as according to their parents’ preference received sedation in a rather generous way after informed consent of their parents according to our department standards. The individual neurophysiological stage of development and the level of parental, as well as children’s stress toward the medical environment were the most important factors for the indication of sedation.

Initially, oral liquid chloralhydrate (Chloralhydrat orale Lösung^®^ 50 mg/mL, 25 mg/kg = 0.5mL/kg) was used in children younger than 6 months, while patients older than 6 months received chlorprothixene (Truxal Saft 0.25%^®^, 1.5 mg/kg p.o.) with no patients kept nil per os. If the sedation was deemed insufficient, midazolam (Dormicum, 0.3 mg/kg) was administered additionally. Sedated children were monitored by pulse oximetry. Physical examination in addition to regular monitoring were carried out by the nursing staff.

### 2.4. Part 3 (Audit after Protocol Change)

According to the results of Part 2, the internal standard operating procedures were adapted after an extensive discussion with all staff involved in Q4/ 2017: children were not sedated by default, but only if doctors, nurses or parents had experienced increased restlessness during previous examinations. Parents were educated in detail, based on the data now available, about the benefit expected from pharmacological sedation. Furthermore, the standard sedative regimen was changed to melatonin (Melatonin-Espara 5 mg < 10 kg bodyweight, 10 mg > kg bodyweight) or, if not sufficient, midazolam (Dormicum, 0.3 mg/kg) for sedation. Patients requiring midazolam sedation were kept nil per os for 6h concerning solid food and 1h concerning clear fluids. As part of the current retrospective analysis, all children who had undergone a nuclear medicine examination at our department between May and November 2020 were evaluated to assess the sedation rate, as well as the rate of unsuccessful exams. Parental satisfaction was recorded via the departmental parent and patient questionnaire issued biannually.

Demographic and exam-related data were extracted from the hospital patient information system (SAP SE, Walldorf, Germany). An overview of the patients’ characteristics is provided in Table 1.

Data were fed into a Microsoft Office Excel sheet (Microsoft Corporation, Redmond, WA, USA), and for further statistical analysis, JUMBO (Java-supported Münsterian biometrical platform) was used.

For the comparison of categorical data-matched pairs, signed-rank Wilcoxon tests were used; for the comparison of parametric data, Student’s t-test was used. A *p*-value < 0.05 was considered significant.

## 3. Results

### 3.1. Part 1

A total of 26 departments performing nuclear medicine studies in Austria were identified via the website of the Austrian Society of Nuclear Medicine (www.ogn.at) (accessed on 1 October 2018). All departments were contacted by email and those who did not respond, additionally by telephone. A total of 15 nuclear medicine departments were willing to provide information (57% response rate). Thereof, forty percent (6/15) did not offer nuclear medical examinations in children. This left nine departments for analysis. Seventy-seven-point-eight percent (7/9) performed DMSA and MAG3 mostly with sedation, partly requiring nil per os (3/7) with anaesthesia participation and monitoring and per individual indication; two departments (22.2%) performed nuclear medicine studies in children only under general anaesthesia.

### 3.2. Part 2

A total of 68 patients undergoing 70 investigations were recruited during the prospective study period (February 2015–May 2015). In the prospective group, twenty-eight-point-five percent of patients (20/70) were sedated by use of chlorprothixene (with only five patients requiring additional medication with midazolam).

In 65% (13/20) of patients where exams were carried out under sedation with chlorprothixene, parents felt that their child was optimally sedated. For 80% (16/20) of parents, the use of a sedative created no additional problems for their children. Ninety-one percent (66/70) of parents mentioned that general anaesthesia (i.e., without parents being present all the time and the child being administered i.v. sedation or airway control) would be unacceptable. For 70% (49/70) of the interviewed parents, the investigation was not stressful, and eighty-six percent (60/70) would agree to perform an investigation under identical conditions again.

Thirty-five percent of patients (7/20) under sedation with chlorprothixene were awake during the examination. In comparison, in the non-sedation group, sixty-seven percent (31/46) of the children did not sleep at all. Thus, the sedative led to a significant decrease in the probability of a non-sleeping child during the examination (*p* = 0.01441, OR 0.260 95% CI 0.7379–0.8925).

According to the nuclear medicine staff, fifty-three percent (37/70) of the patients were calm during the examination, and in twelve percent, restlessness was reported. Motion artefacts were reported in 11.4% (8/70) of all studies, but there was no interference with the interpretation of the results. Seven percent (5/70) of the non-sedated children required additional time due to their restlessness (2× < 10 min; 3× > 10 min). The average age in this group was 4.3 years (median: 3.75 years).

Patients aged over five years (*n* = 22) without sedation were questioned about their feelings concerning the examination using a VAS with face pictograms to differentiate mood. Seventy-seven percent (17/22) felt “good” during the investigation, and seventy-three percent (16/22) would come again. Eighty-two percent (18/22) had the impression that “mom and dad also felt comfortable”.

In addition, three-hundred sixty-five investigations were assessed retrospectively, and the sedation rate was 21%.

The indication for sedation was related to the age of the patient. The indication for sedation was most frequently made in children in the first (59.0% or *n* = 54) and second year of life (51.0% or *n* = 24). Figure 1. Age related sedation rate.

There were no major adverse events in regard to sedation. Agitation as a specific adverse drug reaction was noted only in one (1/102 of all sedated patients during the retrospective and prospective study phase) patient. In 7% (7/102) of patients, the sedative was not sufficient; however, presumptive termination of an investigation was never necessary in sedated patients. On the contrary, in 0.9% (3/333) of the non-sedated patients, the investigation had to be terminated presumptively, in one patient (0.3%) due to agitation of the child and in two patients (0.6%) due to extravasation of the tracer.

### 3.3. Part 3 (Audit after Protocol Change)

After the adaptation of the sedation protocol, we performed 159 nuclear medicine investigations during the study period from May–November 2020. Seventy-point-five percent (12/159) of patients between the ages of one and three years received sedation due to the nuclear medical examination. Seven patients only received melatonin for sedation; 4 patients received midazolam as a sedative; and one patient needed both agents. No examination had to be terminated because of unrestfulness, and no complaint about any problem was noted on the parental or patient side. Figure 2 Sedation rate after audit and protocol change compared to the initial sedation rate before study.

## 4. Discussion

While the Guidelines of the Nuclear Medicine Society are clear about the importance of child friendly examination protocols, there is only a small literature on the assessment of the level of discomfort during nuclear medicine investigations in children [3]. Some studies concerning imaging in paediatric urology, however, aimed at evaluating the level of anxiety and parental satisfaction.

This study showed how a protocol evaluation with a concentrated effort to optimize child friendly examination conditions (indication for sedation, parental information and precise training of staff) led to lower sedation rates while maintaining satisfactory outcomes.

Nelson et al. used a survey to report parental and children’s reactions to paediatric genitourinary diagnostic imaging (GUI). In that study, they compared the experiences of families undergoing a range of GUI tests by a set of questions pertaining to, amongst others, pain, psychological impact and overall satisfaction in 263 children. Compared to renal and bladder ultrasound, DMSA and MAG3 scans showed markedly and significantly higher scores concerning the psychological impact (9.25 vs. 6.2, *p* < 0.001). The age group between one and three years proved to be significantly more anxious as compared to younger or older children [7], supporting our notion that this age might the most problematic and that, as compared to other common exams in urologic conditions, DMSA and MAG3 scans cause relatively more distress.

Kai et al. [6] recommend raising the limit for the need for sedation up to two years after re-evaluating their sedation regimen and using a vacuum mattress for immobilization before the investigation. Based on our results, children between one and two years might be a particularly sensitive age group. Therefore, children between one and three years of age in particular are evaluated at our centre for the need of sedation.

The data from the prospective study, which was carried out during a period when our old sedation scheme was still in use, showed, however, that the children and parents were satisfied with the examination conditions. Seventy-three percent of patients older than five years (16/22) would repeat the investigation if necessary. Interestingly, according to the children, eighty-two percent of parents “felt comfortable” during the exam. This estimation of the older children (>5 years) was exactly in line with what parents in this age group reported, where 84% felt calm and confident.

Although the sedation resulted in a higher rate of sleeping children during the examination, the examinations could be carried out in both groups without any problems or loss of time. Our interpretation of this finding was, that sedation does not itself lead to any improvement in the quality of the examination, and therefore, in order to ensure an improvement in patient safety, sedation might confidently be dispensed with.

On the basis of adequate pre-emptive information of children and parents, as well as trained and motivated nuclear medicine, which help to create a child friendly environment, the sedation rate was reduced from 27.1% to 7.5%. Moreover, there were no unsuccessful examinations documented, and the substances used were altered in favour of a more favourable side-effect profile.

Of the children receiving sedation in the audit group, only 4/11 received benzodiazepines, while most (7/11) only received melatonin: melatonin has emerged relatively recently as a potential alternative sedation in paediatric patients. A meta-analysis, while being limited by the heterogeneous data available, concluded it to be safe and effective [8]. While there are no reports from nuclear medicine exams, its use in other indications, e.g., MRT, is well documented [9]. The effects described in children are not only sedation, but also anxiolysis, as well as anaesthesia, rendering it especially valuable. Relevant side-effects are not documented [10].

The strengths of our study include its prospective evaluation of parental and children satisfaction, the inclusion of many patients and the comparison of our current and past sedation regiments.

In previous studies, especially for the assessment of sedation effects, psychological tools as for example the Groningen Distress Rating Scale were used [11,12]. In order to assess coping with stressful medical events, the Utrecht Coping List is commonly applied, mostly in relation to malignant disease [13]. For our purposes, however, these tools did not seem appropriate. Thus, with the assistance from a psychologist, we designed a set of questions relating better to the individual examination situation present in our nuclear medicine setting.

Our study was limited by the lack of a control group, especially in patients undergoing the investigation under sedation. Consequently, we cannot answer the question whether any sedation is crucial to perform these examinations. Furthermore, we did not use standardized questionnaires, which made our data solitary and non-comparable to other, future studies.

## 5. Conclusions

This analysis demonstrated that a further reduction of sedation is feasible and assures accurate investigations with unimpaired diagnostic quality and patient satisfaction. Additionally, the implementation of the measures taken after a targeted audit herein did not result in a significant expense in time due to the interaction with patients during the exam, and image quality was uncompromised.

By striving to improve and not hold to old standards, protocol adaption (sedation as required, precise parental instructions and training of the staff) and using melatonin in 4.4% of patients further reduced the need for benzodiazepine sedation to 3.1%.

## Figures and Tables

**Figure 1 children-08-00424-f001:**
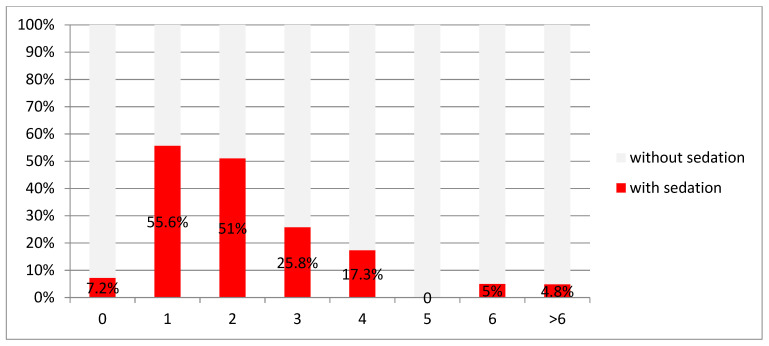
Age-related sedation rate (Part 2: primary data assessment).

**Figure 2 children-08-00424-f002:**
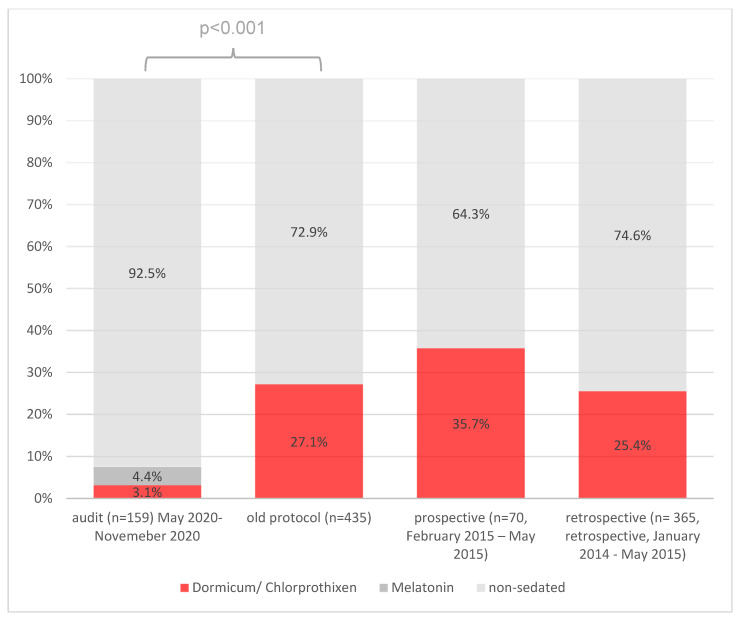
Percentage of sedation rate in nuclear medicine examinations within the retrospective, prospective and current audit analysis. *p* > 0.001 refers to a comparison (chi^2^ test) of sedation rates between the “audit” and “old protocol” groups.

**Table 1 children-08-00424-t001:** Patient characteristics for the new and old protocol (prospective and retrospective cohort).

	Audit (after Protocol Change) (*n* = 159) May 2020–November 2020	Prospective Analysis (*n* = 70) February 2015–May 2015	Retrospective Analysis (*n* = 365) January 2014–May 2015
Sex: male (%)/female (%)	74 (46.5%)/85 (53.5%)	31 (44.2%)/39 (55.8%)	166 (45.4%)/199 (54.6%)
Age mean (range)	24 months (4–68)	25 months (0–169)	21 months (0–212)
DMSA (%)/MAG3 (%)	108(67.9%)/51(32.1%)	45 (64.2%)/25 (35.8%)	220 (60.2%)/145 (39.8%)
Sedation rate (%)	Melatonin 7 (4.4%) Dormicum 5 (3.1%)	Chlorprothixene 20 (28.5%) Dormicum 5 (7.1%)	Chlorprothixene 77 (21%) Dormicum 16 (4.3%)

## Data Availability

Raw data are archived: 10.5281/zenodo.4687654.

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
