# Peer review of "Sedation Rate Reduction in Paediatric Renal Nuclear Medicine Examinations: Consequences of a Targeted Audit"

_children, 2021, doi:10.3390/children8050424_

Round 1
Reviewer 1 Report
The authors evaluated the effect of reduction of sedation rate in pediatric renal nuclear medicine examinations and found that through the application of the most appropriate protocols, staff education, and the use of optimized sedatives, the need for sedation could be reduced whilst maintaining a high patient satisfaction. This is a unique and important study in the field of pediatrics and is potentially interesting. However, I have some concerns as listed below:
- Are MAG3 and DMSA scans performed in the same procedure (requires the same sedation time) in authors’ institute? MAG3 scans (dynamic scintigraphy) take 40-60 minutes for the entire scan, while DMSA (static) scintigraphy should typically take only about 15 minutes to create an image 90-120 minutes after DMSA injection. If the author's institute uses the same procedure in both scans, isn't it important to improve it for sedation rate reduction?
- As suggested, maintaining the quality of the diagnosis without sedation may require a lot of effort, including staff preparation and staff training. Given the burden on medical staff, can the authors reach the same conclusion?
- As described by the authors, evaluation of patients’ satisfaction is difficult. The author should state in the text the content of the questionnaire (at least some examples) used to assess patient satisfaction.
Author Response
Dear Editor, dear Reviewers,
We highly appreciate the chance to further work on our manuscript in order to have it published in your distinguished journal.
The reviewer’s comments proved to be helpful and constructive and we feel to be able to respond accordingly to each of them. Please find a point-by-point response, including information on the changes made to the manuscript below.
Comments are in italic, changes to the manuscript have been implemented in colour.
We would like to express our gratitude to the reviewers for their fair and thoughtful reviews, raising important points yet underestimated in our manuscript.
Are MAG3 and DMSA scans performed in the same procedure (requires the same sedation time) in authors’ institute? MAG3 scans (dynamic scintigraphy) take 40-60 minutes for the entire scan, while DMSA (static) scintigraphy should typically take only about 15 minutes to create an image 90-120 minutes after DMSA injection. If the author's institute uses the same procedure in both scans, isn't it important to improve it for sedation rate reduction?
Thank you for this thoughtful remark; this is an important point to discuss.
In contrast to the guidelines, at our nuclear medicine department DMSA scintigraphy is not only recorded in a planar manner but also SPECT recordings are carried out, which is associated with a slightly longer recording time. Therefore, I would like to briefly describe our examination standards and the different times associated with them.
MAG III: Prior to the examinations, all patients received a standardized intravenous fluid regimen (50ml/h over 3 hours, ELO Paed balanced with 1% glucose ® electrolyte solution by Fresenius Kabi). After at least one hour of standardized hydration, 99mTechnetium labelled MAG3 was injected intravenously. Flow images were acquired at 1 sec/frame for the first 3 minutes, followed by 20 sec/frame images acquired over 27 minutes for F+20 and 1 sec/frame for the first 3 minutes and 20 sec/frame over 17 minutes for F-15 respectively. The basic renogram consisted of a dynamic phase with a minimal duration of 30 min for F+20 and 20 min for F-15, a post micturition image after a mean of 45 minutes and in cases with <50% activity on the post micturition image, a late image 2 hours post injectionem.
DMSA: For DMSA scintigraphy, the infusion was started half an hour before the tracer application and the actual recordings were made after 2 hours. Since we not only perform planar imaging but also SPECT (renal single photon emission computerized tomography (SPECT)) imaging to guarantee better detection of cortical defects, the entire DMSA scintigraphy takes about 45 minutes.
Due to these department standards, the duration of the examination is roughly the same and therefore needs the same sedation time.
A respective adaption in the material and methods part of the manuscript has been added.
The basic renogram consisted of a dynamic phase with a minimal duration of 30 min for F+20, a post micturition image after a mean of 45 minutes and in cases with <50% activity on the post micturition image, a late image 2 hours post injectionem.
For the DMSA scintigraphy we not only do planar imaging but also SPECT (renal single photon emission computerized tomography (SPECT)) imaging to guarantee better detection of cortical defects, therefore the entire DMSA scintigraphy takes about 45 minutes.
As suggested, maintaining the quality of the diagnosis without sedation may require a lot of effort, including staff preparation and staff training. Given the burden on medical staff, can the authors reach the same conclusion?
That is an important and evident point, thank you for bringing it up. We agree with you that in the beginning it is a big change for everyone involved, because you have got used to the circumstance of a sedated child. However, after we saw that sedation did not improve the results or that the same satisfactory results could be obtained without sedation, it was easy to convince everyone involved to the changed protocol. Based on our data and experience, one cannot say that the examinations with sedation would have been better, but the change in the protocol showed that the good results of the patients and parents' satisfaction are not due to the medication but to the child-friendly methods.
As described by the authors, evaluation of patients’ satisfaction is difficult. The author should state in the text the content of the questionnaire (at least some examples) used to assess patient satisfaction.
Thank you for this remark; we clarified this important point in the manuscript.
Patients aged over 5 years were questioned using visual analog scale with face pictograms to differentiate mood. Questionnaire included questions about the wellbeing of the child (example given: How did you feel during the examination? Would you come again?) and parents' assessment of the children’s satisfaction as used in younger children (example given: Do you find the examination situation very stressful? Was your child calm during the examination? Did you feel well looked after?)
Reviewer 2 Report
Good and timely article on a very important issue.
The article is overall well written.
Author Response
We are delighted to receive such feedback. We are pleased that the content of our manuscript also seems to be of great importance to the reviewer. We thank you for the time spent with our manuscript in the context of the review process.
Round 2
Reviewer 1 Report
The authors properly replied to my suggestions, and nicely put those comments in the manuscript. I have no more comments.
Author Response

(The authors gave the same response as above.)
